# Tetra-2,3-Pyrazinoporphyrazines with Externally Appended Pyridine Rings 22 Synthesis, Physicochemical and Photoactivity Studies on In(III) Mono- and Heteropentanuclear Complexes

**DOI:** 10.3390/molecules27030849

**Published:** 2022-01-27

**Authors:** Maria Pia Donzello, Giulia Capobianco, Ida Pettiti, Claudio Ercolani, Pavel A. Stuzhin

**Affiliations:** 1Dipartimento di Chimica, Università di Roma Sapienza, P. le A. Moro 5, I-00185 Rome, Italy; giuliacapobianco.gc16@gmail.com (G.C.); ida.pettiti@uniroma1.it (I.P.); claudio.ercolani@fondazione.uniroma1.it (C.E.); 2Department of Organic Chemistry, Ivanovo State University of Chemistry and Technology, 153460 Ivanovo, Russia; stuzhin@isuct.ru

**Keywords:** mono/pentanuclear In(III) porphyrazine complexes, physicochemical properties, photoactivity studies in PDT

## Abstract

The basic macrocyclic octapyridinotetrapyrazinoporphyrazine In^III^ complex of formula [Py_8_TPyzPzIn(OAc)]·8H_2_O, prepared by reaction of the free ligand [Py_8_TPyzPzH_2_]·2H_2_O with In(OAc)_3_, is a stable-to-air species of which the structure has been studied by its X-ray powder diffraction and mass spectra and characterization operated by IR and UV-visible spectral behavior. The complex has been further examined and proven to be of potential interest for its response as an anticancer agent in the field of photodynamic therapy (PDT), the value of Φ_Δ_ = 0.55 (in DMF) being in the range of 0.4–0.6 at the level of similar phthalocyanine and porphyrazine analogs and qualifying the species as a highly efficient anticancer agent. Planned parallel types of investigation, including their photoactive behaviour in PDT, have been extended to the mononuclear octacation [(2-Mepy)_8_TPyzPzIn(OAc)]^8+^ (salted by iodide ions) and the heteropentanuclear derivatives [(M’Cl_2_)_4_Py_8_TPyzPzIn(OAc)]·xH_2_O (M’ = Pd^II^, x = 8; Pt^II^, x = 1)) and [{(Pd(CBT)_2_)_4_}Py_8_TPyzPzIn(OAc)]·19H_2_O (CBT = *m*-carborane-1-thiolate anion).

## 1. Introduction

Among our extensive studies on macrocyclic porphyrazine metal derivatives often finalized to explore their tendency to behave as active agents in photodynamic therapy (PDT), a well-known anticancer curative modality [1,2,3,4], we previously reported on the mononuclear complexes of formula [TTDPzMX] (TTDPz = tetrakis(thiadiazole)porphyrazinato anion) carrying centrally MX groups (MX = Al^III^Cl, Ga^III^Cl, and In^III^OAc) (OAc = acetate anion) [5], and for the two Al^III^ and Ga^III^ macrocycles relevant data were obtained as to their potential photoactivity in PDT (Φ_Δ_ respectively 0.35 and 0.69) [6]. In recent times, a related type of investigation regarded the mono- and pentanuclear porphyrazine macrocycles of formula [Py_8_TPyzPzMCl]·xH_2_O and [(PdCl_2_)_4_Py_8_TPyzPzMCl]·xH_2_O (Py_8_TPyzPz = tetrakis-2,3-[5,6-di(2-pyridyl)-pyrazino]porphyrazinato dianion; M = Al^III^, Ga^III^) [7] (Figure 1), all data cited appearing well compatible with those known for similar phthalocyanine and porphyrin analogs. Based on these data, it was found interesting to add to these findings those regarding a series of In^III^ related porphyrazine complexes, this allowing a comparison on the heavy metal effect along the triad of the Al^III^, Ga^III^, In^III^ complexes and on the role played by the target macrocycle as a potential therapeutic agent in PDT. From the basic complex [Py_8_TPyzPzIn(OAc)], the related octacation [(2-MePy)_8_TPyzPzIn(OAc)]^8+^ (as the iodide salt) and the hetero-pentanuclear derivatives [(M’Cl_2_)_4_Py_8_TPyzPzIn(OAc)] (M’ = Pd^II^, Pt^II^) and [{(Pd(CBT)_2_)_4_}Py_8_TPyzPzIn(OAc)] (CBT = *m*-carborane-1-thiolate anion, C_2_H_11_B_10_S) were also prepared, all of them obtained as hydrated species, mainly recalling the structural features of the related Al^III^ and Ga^III^ complexes (Figure 1). In terms of potential photoactivity as anticancer agents in PDT, interesting data were determined for the macrocycles [Py_8_TPyzPzIn(OAc)] (Φ_Δ_ = 0.55) and the two pentanuclear species [(M’Cl_2_)_4_Py_8_TPyzPzIn(OAc)] (M’ = Pd^II^, Pt^II^) (Φ_Δ_, respectively, 0.36 and 0.46), with values close or well within the range observed for most phthalocyanine and porphyrazine analogs (Φ_Δ_ = 0.4–0.6). A detailed physicochemical characterization was performed on all the species by elemental analysis and X-ray, mass, IR and UV-visible spectral measurements.

## 2. Experimental Section

**Solvents and Reagents.** Solvents were from Carlo Erba, Fluka, or Aldrich; dimethyl sulfoxide (DMSO) was freshly distilled on CaH_2_ before use; pyridine was dried by refluxing over CaH_2_. Carborane-1-thiol (HSC_2_B_10_H_11_, CBTH) was purchased from Katchem ltd and used as received. [(C_6_H_5_CN)_2_PtCl_2_] was prepared by a procedure similar to that used for the synthesis of [(C_6_H_5_CN)_2_PdCl_2_] [8]. The macrocycle [Py_8_TPyzPzH_2_]·2H_2_O was prepared as previously reported [9].

**[Py_8_TPyzPzIn(OAc)]∙8H_2_O**. [Py_8_TPyzPzH_2_]∙2H_2_O (100.9 mg, 0.086 mmol) and In(OAc)_3_ (48.9 mg, 0.167 mmol) (molar ratio 1:2) were suspended in glacial CH_3_COOH (5 mL) and the mixture was heated under stirring at 135 °C for 2 h. After cooling to room temperature and the addition of H_2_O (5 mL), the suspension was kept in a refrigerator overnight. After centrifugation, the separated dark green solid was washed with H_2_O until neutrality of the washings and was brought to constant weight under vacuum (10^−2^ mmHg) (48.0 mg; yield 39.0%). Calcd for [Py_8_TPyzPzIn(OAc)]∙8H_2_O, C_66_H_51_InN_24_O_10_: C, 54.48; H, 3.53; N, 23.10; In, 7.89. Found: C, 54.63; H, 2.86; N, 22.72; In, 8.30 %. IR (cm^−1^): 3600–3300 broad, 1624 (vw), 1574 (m), 1557 (w), 1534 (w), 1483 (w), 1463 (w), 1423 (vw), 1350 (m-s), 1283 (vw), 1235 (vs), 1189 (w-m), 1144 (vw), 1109 (m-s), 986 (w-m), 950 (m-s), 846 (vw), 823 (vw), 781 (w-m), 743 (w), 701 (s), 653 (w-m), 625 (w), 551 (w), 469 (vw), 436 (vw), and 402 (vvw).

**[(2-Mepy****)_8_TPyzPzIn(OAc)](I)_8_****·H_2_****O**. [Py_8_TPyzPzIn(OAc)]∙8H_2_O (22.5 mg, 0.0155 mmol) and CH_3_I (0.20 mL, 3.22 mmol) were added to DMF (1 mL) and the mixture was kept under stirring at room temperature for 42 h. After evaporation of the excess of CH_3_I in air at room temperature, the solid present in the mixture was separated and discarded and the liquid was kept overnight in a refrigerator after addition of benzene (4 mL). The dark green solid material formed, meanwhile, was separated by centrifugation, washed with benzene and ethyl ether and brought to constant weight under vacuum (10^−2^ mm Hg) (18.8 mg, yield 50%). Calcd for [(2-Mepy)_8_TPyzPzIn(OAc)](I)_8_·H_2_O C_74_H_61_I_8_InN_24_O_3_: C, 36.33; H, 2.43; N, 13.74; In, 4.69. Found: C, 36.59; H, 3.30; N, 13.43; In, 4.41 %. IR (cm^−1^): 3500–3300 broad, 3033 (vw), 2991 (vw), 2941 (vvw), 2924 (vvw), 1732 (vw), 1640 (m-s), 1611 (s), 1569 (m-s), 1542 (m), 1504 (w), 1473 (vw), 1450 (w), 1402 (vw), 1351 (m), 1270 (m), 1236 (s), 1176 (m), 1146 (m-s), 1109 (w-m), 1088 (w-m), 989 (w), 937 (w-m), 845 (vw), 822 (vw), 768 (w), 741 (w), 711 (w), 686 (w-m), 648 (w), 623 (w), 563 (vw), 521 (vvw), and 435 (vw). 

**[(PdCl_2_)_4_Py_8_TPyzPzIn(OAc)]∙8H_2_O**. [Py_8_TPyzPzIn(OAc)]∙8H_2_O (20.7 mg; 0.014 mmoli) and (C_6_H_5_CN)_2_PdCl_2_ (26.3 mg (0.069 mmoli) (molar ratio 1:4.8) were introduced in a flask (10 mL) containing CHCl_3_ (5 mL) and the mixture was kept under stirring a room temperature for 24 h and then heated at 50 °C for 48 h. After cooling and centrifugation, the separated dark green solid was washed with acetone and brought to constant weight under vacuum (10^−2^ mmHg; 25.0 mg, yield 81.0%). Calcd for [(PdCl_2_)_4_Py_8_TPyzPzIn(OAc)]∙8H_2_O, C_66_H_51_Cl_8_InN_24_O_10_Pd_4_: C, 36.62; H, 2.37; N, 15.53; In, 5.30; Pd, 19.67%. Found: C, 37.12; H, 2.86; N, 14.92; In, 4.79; Pd, 20.17%. IR (cm^−1^): 3600–3300 broad, 1735 (vw), 1618 (m), 1590 (m), 1557 (w-m), 1447 (w-m), 1352 (m), 1227 (s), 1235 (vs), 1182 (w), 1153 (w-m), 1115 (w), 1028 (vw), 988 (vw), 952 (w), 848 (vw), 770 (w), 703 (m), 651 (vw), 553 (vw), 434 (vw), 337 (w-m, ν_Pd-Cl_), 289 (vvw). 

**[(PtCl_2_)_4_Py_8_TPyzPzIn(OAc)]****∙H_2_O**. [Py_8_TPyzPzIn(OAc)]∙8H_2_O (20.0 mg, 0.0137 mmoli) and e (C_6_H_5_CN)_2_PtCl_2_ (32.3 mg, 0.0685 mmoli) (molar ratio 1:5) were introduced in a flask (10 mL) containing CHCl_3_ (5 mL) and the mixture was kept under stirring at 65 °C for five days. After cooling and centrifugation, the separated dark green solid was washed abundantly with acetone and brought to constant weight under vacuum (10^−2^ mmHg) (18.20 mg; yield 55.5%). Calcd for [(PtCl_2_)_4_Py_8_TPyzPzIn(OAc)]∙H_2_O, C_66_H_37_InCl_8_N_24_O_3_Pt_4_: C, 33.13; H, 1.56; N, 14.05; In, 4.80; Pt, 32.61%. Found: C, 33.93; H, 2.08; N, 13.88; In, 4.89; Pt, 32.26%. IR (cm^−1^): 3600–3300 broad, 1734 (w), 1638 (m), 1592 (m), 1553 (m), 1475 (m), 1427 (vw), 1408 (vw), 1350 (w-m), 1310 (vw), 1236 (vw), 1235 (vs), 1184 (w-m), 1149 (m), 1113 (w-m), 1086 (w-m), 1040 (vw), 988 (vw), 950 (w-m), 846 (vw), 770 (w), 752 (w), 703 (ms), 651 (w), 624 (w), 551 (w), 436 (w), 339 (w, ν_Pt-Cl_), 298 (vw). 

**[{Pd(CBT)_2_}_4_Py_8_TPyzPzIn(OAc)]∙19H_2_O (CBT = *m*-carborane-1-thiolate anion)**. [(PdCl_2_)_4_Py_8_TPyzPzIn(OAc)]∙8H_2_O (14.8 mg, 0.0068 mmoli) and *m*-carborane-1-thiol (17 mg, 0.096 mmol; molar ratio 1:14) were introduced in a flask (10 mL) containing CH_3_CN (5 mL) and the mixture was heated under reflux with stirring at 90 °C. After cooling and centrifugation, the dark green solid was repeatedly washed with CH_3_CN at room temperature and then kept at 50 °C in the same solvent for 30′. After cooling, the isolated solid was brought to constant weight under vacuum (10^−2^ mmHg) (20.1 mg; yield 85.2%). Calcd for [{Pd(CBT)_2_}_4_Py_8_TPyzPzIn(OAc)]∙19H_2_O, C_82_H_151_B_80_InN_24_O_21_Pd_4_S_8_: C, 28.37; H, 4.38; N, 9.68; In, 3.31; Pd, 12.26%. Found: C, 27.77; H, 4.01; N, 9.18; In, 3.10; Pd, 12.53%. IR (cm^−1^): 3600–3300 broad, 3040 (vw), 2586 (vs), 2335 (vvw), 1728 (vw), 1624 (vw), 1350 (vw), 1145 (vw), 1124 (vw), 1069 (m), 772 (w-m), and 473 (w). 

**Singlet Oxygen Quantum Yield Measurements.** Measurements of singlet oxygen quantum yield (Φ_Δ_) were carried in DMF by an absolute method using 1,3-diphenylisobenzofuran (DPBF) as the scavenger of singlet oxygen (^1^O_2_), as previously reported [6]. Solutions of the complexes (ca. 10^−6^–10^−5^ M) and DPBF (ca. 5 × 10^−5^ M) in DMF were irradiated in a 10-mm path length quartz cell with monochromatic light (Premier LC Lasers/HG Lens, Global Laser). The irradiation wavelength (λ_irr_ = 670 nm) was close to the maximum of the Q-band absorption peaks for each compound. The light intensity was set to 0.300 mW and the value accurately measured with a radiometer (ILT 1400A/SEL100/F/QNDS2, International Light Technologies). The decay of DPBF absorption at 414 nm (ε^DPBF^ = 2.3 × 10^4^ mol^−1^ L cm^−1^) was monitored at 20 °C by a Varian Cary 50 Scan UV-visible spectrophotometer. The Φ_Δ_ values were calculated from Stern–Volmer plots on the basis of Equation (1): (1)$$\frac{1}{\Phi_{DPBF}}=\frac{1}{\Phi_{\Delta }}+\frac{k_{d}}{k_{r}}\frac{1}{\Phi_{\Delta }}\frac{1}{\left[DPBF\right]}$$
where Φ_DPBF_ is the quantum yield of the photoreaction, k*_d_* is the decay rate constant of ^1^O_2_ in the solvent, and k*_r_* is the rate constant for the reaction of DPBF with ^1^O_2_. The 1/Φ_Δ_ values were obtained as the intercept of each linear plot (1/Φ_DPBF_ versus 1/[DPBF]).

**Other physicochemical measurements**. IR spectra were recorded on a Varian FT-IR 660 instrument in the range of 4000–250 cm^−1^ (KBr pellets or nujol mulls between CsI disks). UV-visible solution spectra of the synthesized compounds were recorded with a Varian Cary 5E spectrometer using quartz cuvettes (1 cm). Elemental analyses for C, H, N, and S were provided by the “Servizio di Microanalisi” at the Dipartimento di Chimica, Università “La Sapienza” (Rome), on an EA 1110 CHNS-O instrument. The ICP PLASMA In, Pd, and Pt analyses were performed on a Varian Vista MPX CCD simultaneous ICP-OES. X-ray powder diffraction spectra were run in the interval 500–5000 (2 ϑ/°C) on a Philips PW1029 instrument interfaced with a computer (Software APD Philips), using a CuKα radiation. Mass spectra were recorded on a Shimadzu Biotech Axima Confidence spectrometer operating in MALDI-TOF modality (Collective Usage Center of Ivanovo State University of Chemical Technology supported by the Ministry of Science and Higher Education of Russia, grant No. 075-15-2021-671).

## 3. Results and Discussion


**[**
**Py_8_TPyzPzIn(OAc)]∙8H_2_O and its corresponding octacation [(2-Mepy)_8_TPyzPzIn(OAc)]^8+^ (salted by I^−^ ions).**


(a)*General properties.* The basic mononuclear In^III^ complex [Py_8_TPyzPzIn(OAc)]∙8H_2_O, prepared by heating in glacial CH_3_COOH the reactants [Py_8_TPyzPzH_2_]·2H_2_O and In(OAc)_3_ (Section 2), is generally obtained in good yield as a brilliant green solid. The X-ray powder spectrum of the complex (Figure 2A), indicative of a partial crystalline character, is closely approaching the X-ray powder of the spectra of the related macrocyclic analogs centrally carrying Al^III^Cl and Ga^III^Cl units (Figure 2B,C) [7]. As established for the parallel series of tetrakis(thiadiazole)porphyrazines [TTDzPzMX] (M = Al^III^, Ga^III^; X = Cl/M = In^III^, X = OAc), [5] schematically shown in Figure 3, and supported by the molecular arrangement of the corresponding Al^III^ and Ga^III^ species both elucidated by X-ray work [5] and by the findings for other similar tetrapyrrolic macrocycles given in ref. [7] (Table 1), providing for all of them (M = Al^III^, Ga^III^) a distance M-*C*_t_ of 0.3–0.4 Å, the In^III^ center in the present basic macrocycle is given as axially positioned and residing out of the *C*_t_ of the inner N_4_ coordination site probably only very slightly exceeding the value of 0.4 Å. The water present in the complex, most likely involved in different forms of hydrogen bonds with the numerous N atoms of the macrocycle, could be eliminated by heating it under vacuum (100 °C, 10^−2^ mmHg); nevertheless, rehydration took place at least partially by exposition of samples to the air. Only traces of water were present in the corresponding macrocyclic salt-like species, most likely explained by the fact that external N atoms once positively charged are no longer available for water attraction.

Useful information about the molecular formula of the basic In^III^ complex is provided by its mass MALDI-TOF spectrum (Figure 4). Peak positions at *m*/*z* = 1252.5 and 1287.5 were assigned, respectively, to the molecular fragments [Py_8_TPyzPzIn] and [Py_8_TPyzPzInCl], the latter formed clearly by the substitution of the OAc^−^ group with Cl^−^ caused by the utilized solvent (CHCl_3_). In addition, a very low signal positioned at m/z 2391 was also observed as reasonably assigned to the formation of a mononuclear dimeric species [{Py_8_TPyzPz}_2_In] showing the metal center bridging two macrocyclic units. Noteworthy, a low intensity signal of the molecular peak at *m*/*z* = 1311.7 present in the bottom spectrum identified the species [Py_8_TPyzPzIn(OAc)]. It is noteworthy that, in the MALDI-TOF spectrum recorded using solution in CH_3_CN, the molecular ion peak corresponding to the species with axial chloride is absent, along with the stable cation [Py_8_TPyzPzIn]^+^ peak of the bis(acetate) complex [Py_8_TPyzPzIn(OAc)_2_]^+^, could be seen (see Appendix A and related discussion in Supporting Information).

The newly reported related porphyrazine octacation [(2-Mepy)_8_TPyzPzIn(OAc)]^8+^ schematically shown in Figure 5 is present in the corresponding iodide salt formulated as [(2-Mepy)_8_TPyzPzIn(OAc)](I)_8_·H_2_O, which was obtained as a dark green solid by heating the basic In^III^ neutral complex in the presence of CH_3_I in DMF (Section 2) and proved to be an amorphous material by its X-ray powder diffraction spectrum (Appendix A). The restricted presence of water in the salted species is reasonably explained by the occurring engagement of the external N atoms in the methylation process and their associated quaternization. 

(b)*IR and UV-visible spectral behavior.* The IR spectrum of the present basic In^III^ complex shows, in addition to broad absorptions observed in the region just above 3000 cm^−1^ as due to the O-H stretching of the present water molecules, the typical absorptions of the Py_8_TPyzPz porphyrazine macrocycle present in the region 1800–280 cm^−1^ (Figure 6, top), including the doublet at 986 and 950 cm^−1^, closely recalling those observed previously for the Al^III^ and Ga^III^ analogs and the complexes [Py_8_TPyzPzM] (M = bivalent metal center; ref. [7] (Table 1)). In the spectrum of the octacationic-related macrocycle (Figure 5, bottom), the essential skeletal absorptions of the macrocycle were kept in unchanged positions, with partly superimposed new peaks assigned to the presence of bending vibrations of the new CH_3_ groups and changes determined by the introduced N^+^ charged centers in the pyridine rings.

**Figure 6 molecules-27-00849-f006:**
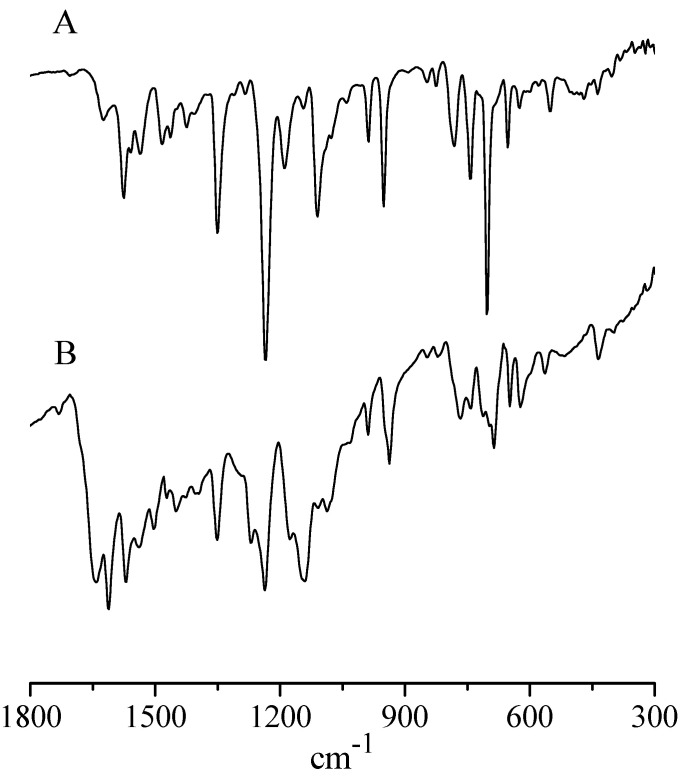
IR spectra of [Py_8_TPyzPzIn(OAc)]·8H_2_O (**A**) and [(2-Mepy)_8_TPyzPzIn(OAc)](I)_8_·H_2_O (**B**). in the range 1800–300 cm^−1^.

The complex [TPyzPzIn(OAc)]∙8H_2_O, completely insoluble in water and very poorly soluble in the nondonor solvents THF, acetone, CH_2_Cl_2_ and CHCl_3_, dissolves a little better in DMSO, DMF, CH_3_CN and pyridine (c ~ 10^−5^ M). The iodide salt of its corresponding octacationic macrocycle, i.e., [(2-Mepy)_8_TPyzPzIn(OAc)](I)_8_·H_2_O is very scarcely soluble in the nonaqueous low polar solvents (CHCl_3_, THF, CH_2_Cl_2_) and better soluble in CH_3_CN, THF, DMSO, and in H_2_O. The UV-visible solution spectra of the neutral species (data in Table 1) taken in CH_3_CN and CHCl_3_ (Appendix A) indicate the presence of the complex in its pure monomeric form with exclusive narrow intense absorptions in the Soret (370–380 nm) and Q-band region (660–670 nm), both assigned as π-π* transitions. Differently, the initial UV-visible spectrum of the complex shows in DMF (Figure 7), in addition to the presence of the Soret and Q-band peaks, an additional absorption of much lower intensity positioned immediately above 700 nm (peak at ca. 740 nm), which increases in intensity along with the time (4h), this effect paralleled by a concomitant increase of a broad absorption of an initial even lower intensity present in the region 500–600 nm. Both these absorptions, most likely indicative of the presence of some form of aggregation, moved in intensity in opposite direction with respect to that of the Soret and Q-band peaks, the spectra evidencing clean isosbestic points. These findings encouraged the hypothesis of an aggregation implying a monomer to dimer process. Moreover, a parallel evolution of the spectrum is observed in DMSO and pyridine (Appendix A).

Interestingly, a systematic bathochromic shift of ca. 10–15 nm was observed by comparing the Q-band position (660–670 nm) for the present In^III^ complex in its monomeric form in the different solvents (Table 1) with that of the respective analogs [Py_8_TPyzPzMCl] and [Py_8_TPyzPzMOH] (M = Al^III^ and Ga^III^) [7]. In this respect, these overall findings also closely recall what was observed for the parallel known triad of [TTDPzMX] complexes (M = In^III^, X = OAc^−^; Al^III^, Ga^III^; X = Cl^−^, OH^−^) [5].

Of remarkable interest is the ensemble of the UV-visible spectral behavior observed for the octacationic macrocyclic complex (as its iodide salt) in the nonaqueous solvents CH_3_CN, DMF, DMSO, H_2_O and py (quantitative data in Table 1). Two main aspects need to be evidenced when observing the spectra run immediately after dissolution of the corresponding salt species in both CH_3_CN and DMF reported in the region 300–900 nm (Figure 8). Both spectra show typical absorptions in the Soret and Q-band region located, respectively, at 355–360 nm and at 670–680 nm, as expected for a porphyrazine macrocyclic unit, with concomitant comparatively intense broad absorptions with the highest intensity present at ca. 550 and above 700 nm. These two latter combined absorptions are certainly indicative of the presence of a heavily more intense form of aggregation from that evidenced for the corresponding neutral species (Figure 6). The same form of aggregation becomes even more evident in DMSO and literally dominant in H_2_O (Figure 9). Similar behavior was observed in pyridine (Appendix A). 

**[(PdCl_2_)_4_Py_8_TPyzPzIn(OAc)]∙8H_2_O and [(PtCl_2_)_4_Py_8_TPyzPzIn(OAc)]∙H_2_O.** Procedures for the synthesis of these two pentanuclear macrocycles as hydrated species, developed by reaction of the mononuclear complex [Py_8_TPyzPzIn(OAc)]∙8H_2_O with [(C_6_H_5_CN)_2_PdCl_2_] or [(C_6_H_5_CN)_2_PtCl_2_] (Section 2), led to the formation of amorphous green brilliant solids, as proven by X-ray powder spectra (Appendix A). 

The IR spectra of both macrocycles in the region 1800–300 cm^−1^ (Figure 10) recall aspects mainly observed for the respective mononuclear species, a well distinct spectral feature being instead for them the presence of the absorption at ca. 340 cm^−1^ assigned, as due to ν_(M-Cl)_ (M = Pd^II^, Pt^II^)_._ Worth notice is also, for both species, the lowered intensity reproducibly observed in the region 1000–950 cm^−1^ of the absorption positioned at 995 cm^−1^ as compared to that present at ca. 950 cm^−1^, both appearing for the mononuclear species of more comparable relative intensity (Figure 5), a difference which can be taken as indicative of the change mono- to pentanuclear species, these findings recalling those occurring for the homo/heteropentanuclear macrocyclic analogs carrying centrally bivalent [10,11] and tervalent metal centers [7]. 

The present two hydrated pentanuclear macrocycles [(MCl_2_)_4_Py_8_TPyzPzIn(OAc)]·xH_2_O (M = Pd^II^, Pt^II^) assigned a structure similar to that reported in Figure 1 for the related Al^III^ and Ga^III^ corresponding species and having peripherally coordination sites of the type represented schematically in Figure 1 involving either PdCl_2_ or PtCl_2_ units ar_e_ less soluble in DMSO and DMF than the corresponding neutral mononuclear In^III^ complex. Their initial UV-visible spectra in the two solvents, similarly to what observed for the respective mononuclear macrocycle, indicate the occurrence of some form of aggregation more heavily increasing with the time (ca. 24 h), as evident for the spectra taken for both species in DMSO (Figure 11) and in DMF (Appendix A). 

The reproducible bathochromic shift observed for the process mono- to pentanuclear species (2–6 nm in DMSO and DMF, 10 nm in CH_3_CN, nul in pyridine; data in Table 1) indicates that peripheral binding of the PdCl_2_ and PtCl_2_ units only scarcely increments the electron-deficient character of the central macrocycle, parallel effects previously observed also for their Pd^II^ and Pt^II^ homo/heteropentanuclear analogs [10,11,12,13]. Moreover, prolonged contact with pyridine determines the external release of the MX_2_ units and complete formation of the related mononuclear In^III^ species, in line with previous findings for the previously studied pentanuclear macrocycles [(PdCl_2_)_4_Py_8_TPyzPzM] (M = bivalent metal center) [10,11].

**[{Pd(CBT)_2_}_4_Py_8_TPyzPzIn(OAc)]·19H_2_O (CBT = *m-*carborane-1-thiolate anion)**. In the area of anticancer BNCT (boron neutron capture therapy), only two “low molecular weight” curative agents have so far been used in clinical trials, i.e., sodium borocaptate (BSH, Na_2_B_12_H_11_SH) and the dihydroxyboryl derivative of phenylalanine (BPA), others being of only recently proposed intent or under evaluation [14] (including carboranyl porphyrins and porphyrazines) [15]. Since, in this area, we recently reported on the hypothesis of potentialities in BNCT of two low molecular weight species, i.e., *cis*-[(bipy)Pd(CBT)_2_] and t*rans*-[(py)_2_Pd(CBT)_2_] (CBT = *m*-carborane-1-thiolate anion) [16], and also on the macrocyclic porphyrazine complexes of formula [{Pd(CBT)_2_}_4_Py_8_PyzPzM]·xH_2_O (M = Mg^II^(H_2_O), Zn^II^, Pd^II^) [17] specifically thought as a bimodal PDT/BNCT active species, added here is the synthesis and physicochemical characterization of the complex [{Pd(CBT)_2_}_4_Py_8_TPyzPzIn(Oac)]·19H_2_O of the proposed structure shown in Figure 12. 

This heteropentanuclear species is obtained as a light green substantially amorphous (Appendix A), highly hydrated solid material by reaction of the parent pentanuclear complex [(PdCl_2_)_4_Py_8_TPyzPzIn(OAc)]·8H_2_O with *m-*carborane-1-thiol (Section 2), this implying full peripheral substitutions of Cl^−^ by CBT^−^ anions on the external Pd^II^ centers. Its IR spectrum (Figure 13) shows in the range 3600–3200 cm^−1^ a broad absorption due to the O-H stretching of water and, in the adjacent region, the low intensity peak at 3042 cm^−1^ and the much more intense peak at 2588 cm^−1^ assigned, respectively, to the stretching of the C-H and B-H bonds internal to the *m-*carborane-1-thiolate anion. As expected, complete absence was observed in the region 400–300 cm^−1^ of the peak due to ν_(Pd-Cl)_ presence in the spectrum of the related starting species. 

The present pentanuclear CBT derivative, insoluble in H_2_O and in the nonaqueous solvents acetone, THF and CHCl_3_, dissolves better (c ~ 10^−5^ M) in DMSO, DMF and pyridine. Its UV-visible spectra in these solvents (Figure 14) exhibit features, in terms of peaks due to ligand-centered π-π* transitions, recalling those shown by the above reported In^III^-related mono/pentanuclear macrocycles. The observed intense broad peak present in the region 400–500 nm, not observed in the spectrum of the complex [(PdCl_2_)_4_Py_8_TPyzPzIn(OAc)] from which the CBT derivative is generated, nor in that of the related In^III^ mononuclear species is interesting. The same characteristic absorption is instead systematically present in the UV-visible spectra of the previously reported homo/heteropentanuclear analogs [{Pd(CBT_2_)}_4_Py_8_TPyzPzM] [17] also carrying externally Pd(CBT)_2_ units with the CBT groups being necessarily in a vicinal position, i.e., *cis-*arranged. From these data, it clearly appeared that the observed 400–500 nm absorption was in intimate relationship with the presence in the macrocycle of the Pd(CBT)_2_ units. Of remarkable additional importance is that in recent work conducted on the two low molecular weight CBT derivatives i.e., *cis*-[(bipy)Pd(CBT)_2_] and *trans*-[py_2_Pd(CBT)_2_], both of the elucidated structures [16], only the cis structure exhibit the same type of absorption in the region 400–500 nm, which has also been observed in the UV-visible spectra of all pertinent macrocycles cited above, [17] whereat it is definitely absent in the spectrum of the *trans*-species. This clearly means that the origin of the absorption is intimately connected with the presence of two *cis*-arranged CBT groups in M(CBT)_2_ units, either in macrocyclic species or in low molecular weight compounds, all the exposed findings supported by detailed theoretical DFT and TDDFT calculations [17].

The present highly hydrated CBT derivative has been observed via its UV-visible spectral behavior in DMSO and DMF (Figure 14, up and center) to remain practically unchanged during 24 h, some persistent aggregation being present as indicated by the absorption observed in the region above 750 nm. In pyridine solution (Figure 14, bottom), during the same time, progressive decreasing of the 400–500 nm absorption was observed, known to be associated, as previously observed for the CBT analogs, [17] to progressive releasing of the external Pd(CBT)_2_ units with concomitant formation of the related mononuclear In^III^ macrocycle. 

**Singlet Oxygen Quantum Yield Studies**. Photodynamic therapy (PDT) is an anticancer treatment based on the use of the components light and dioxygen (^3^O_2_); a third component is a photosensitizer, possibly a macrocyclic tetrapyrrolic metal derivative, able to absorb energy in the range 600–850 nm undergoing excitation from the ground state S_0_ to the triplet state T_1_ with a high quantum yield, and also on having an adequate T_1_ energy and lifetime for a proper energy transfer to dioxygen for the process ^3^O_2_ to ^1^O_2_ to occur, ^1^O_2_ operating as the main cytotoxic agent responsible for the therapeutic effect [1,2,3,4,18,19]. The type of metal center in the macrocycle can strongly influence this process and, in this regard, it is known that incorporation of closed-shell metal ions such as Zn^II^, Mg^II^, Al^III^, Ga^III^, and Si^IV^ give the complexes desirable photophysical properties in terms of high triplet quantum yields and long triplet lifetimes, which are essential for an efficient photosensitization process, as do, in some cases, open shell diamagnetic d^8^ metal centers, such as Pd^II^ and Pt^II^. Among tetrapyrrolic macrocycles carrying centrally tervalent metal ions, the most popular is the Al-photosensitizer PcAlS_mix_ (Photosens^®^), a mixture of Al^III^ hydroxide phthalocyanines with sulphonated side-groups, which is nowadays approved in Russia for various anticancer treatments [20,21]; Ga^III^ and In^III^ phthalocyanine complexes have also received increasing attention as photosensitizers in PDT [22,23,24,25].

In the present work, results are presented on the photosensitizing activity for the generation of singlet oxygen, of the above reported mono- and heteropentanuclear In^III^ complexes. The singlet oxygen quantum yields values (Φ_Δ_) (Table 2) were obtained in pure DMF with a macrocycle concentration of ca. 10^−6^–10^−5^ M. The procedure described in the Section 2 was based on an absolute method, using a laser source at 670 nm close to the maximum of the Q-band absorption peaks of the examined species. Solutions of both mono- and pentanuclear complexes were found stable under laser irradiation during the experiments. Data listed in Table 2 of the Φ_Δ_ values for the related derivatives [Py_8_TPyzPzMCl] (M = Al^III^ and Ga^III^) were obtained in preacidified DMF ([HCl ≅ 1 × 10^−4^ M) due to the instability of these species in DMF [7].

Figure 15 shows the UV-visible spectral data (A) and the relative Stern-Volmer analysis (B, Equation (1) in the Section 2) obtained in a typical experiment carried out with the mononuclear In^III^ derivative [Py_8_TPyzPzIn(OAc)] and used to calculate the singlet oxygen quantum yield value (Φ_Δ_) of the complex. Figure 15A and the inset of Figure 15B illustrate the absorption decay recorded at 414 nm for the ^1^O_2_ scavenger, DPBF, during irradiation, as well as the stability of the complex. 

The Φ_Δ_ value of [Py_8_TPyzPzIn(OAc)]·8H_2_O (0.55) falls in the range of the values 0.4–0.6 previously reported for a number of Zn^II^ phthalocyanines and porphyrazines [26,27,28,29,30,31,32], including porphyrazine macrocycles studied by our group [10,11,12,13,17,33], and qualifies this compound as a highly efficient photosensitizer for the generation of singlet oxygen. Slightly lower values were obtained for the respective pentanuclear complexes [(MCl_2_)_4_[Py_8_TPyzPzIn(OAc)]·xH_2_O (M = Pd^II^, Pt^II^). The values previously obtained in acidified DMF for the mononuclear complexes [Py_8_TPyzPzMCl] (M = Al^III^, Ga^III^; 0.24 and 0.68) agree with the “heavy atom effect”, which enhances the triplet excited state quantum yield for Ga^III^ with respect to Al^III^. Indeed, the introduction of a heavier metal ion into a porphyrazine macrocycle increased the rate of *intersystem crossing* via enhancement of spin-orbit coupling, this favouring the formation of a triplet state T_1_ with adequate energy and lifetime to allow a proper energy transfer to dioxygen for the process ^3^O_2_ to ^1^O_2_. In this regard, although not higher than that for the Ga^III^ analog, appreciable is the observed value for the mononuclear In^III^ species. Irrelevant Φ_Δ_ values were obtained in pure or acidified DMF (0.0–0.1) for the water soluble salt-like macrocycle [(2-Mepy)_8_TPyzPzIn(OAc)](I)_8_·H_2_O and as well for the heteropentanuclear CBT derivative [{Pd(CBT)_2_}_4_Py_8_TPyzPzIn(OAc)]∙19H_2_O, both of these species highly involved, as shown by their UV-visible spectral behavior, in heavy forms of aggregation. Further studies are planned in an attempt to overcome this aspect and allow them to be examined as potential PDT or, tentatively, bimodal PDT/BNCT anticancer agents, eventually using intracellular transport by liposomes as operated by other porphyrazines [34].

## 4. Conclusions

Reaction of the macrocyclic porphyrazine free ligand [Py_8_TPyzPzH_2_]·2H_2_O with In(OAc)_3_ has allowed isolation of the new stable-to-air In^III^ complex [Py_8_TPyzPzIn(OAc)]·8H_2_O, from which a) the iodide salt of its corresponding octacation [(2-Mepy)_8_TPyzPzIn(OAc)](I)_8_·H_2_O, and b) the heteropentanuclear derivatives [(MCl_2_)_4_Py_8_TPyzPzIn(OAc)]·xH_2_O (M = Pd^II^, Pt^II^) and [{Pd(CBT)_2_}_4_Py_8_TPyzPzIn(OAc)]∙19H_2_O (CBT = *m-*carborane-1-thiolate anion) have been prepared. Characterization of all the new macrocyclic complexes has been accomplished by physicochemical exploration mainly focusing on X-ray powder and mass spectra, IR and UV-visible spectral behavior. By proceeding in a comparison with previous results obtained on the parent Al^III^ and Ga^III^ analogs and on the triad of the parallel macrocyclic tetrakis(thiadiazol)porphyrazines [TTDzPzMX] (M = Al^III^, Ga^III^; X = Cl / M = In, X = OAc), the final target has been to provide results that can express for all the new species their curative potentialities in terms of their behaviour as anticancer agents, mainly in the area of photodynamic therapy (PDT), results proving their response being prevalently in the order Ga^III^ > In^III^ > Al^III^, with a value of Φ_Δ_ = 0.55 for the basic In^III^ complex [Py_8_TPyzPzIn(OAc)]·8H_2_O of relevant interest, values of the related heteropentanuclear complexes also being interesting. 

## Figures and Tables

**Figure 1 molecules-27-00849-f001:**
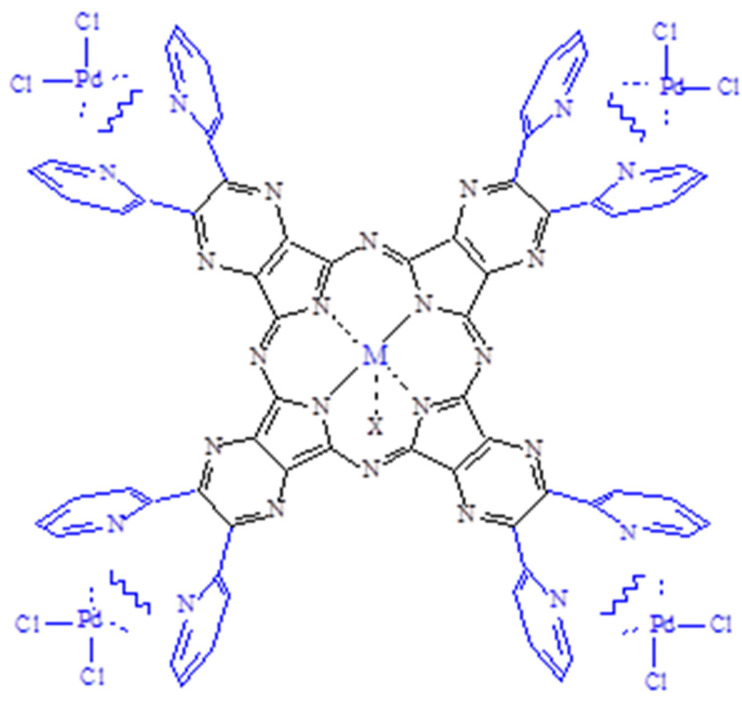
Schematic view of the mononuclear and heteropentanuclear macrocycles [Py_8_TPyzPzMCl] and [(PdCl_2_)_4_Py_8_TPyzPzMCl)] implying external “py-py” coordination (M = Al^III^, Ga^III^; X = Cl) (water neglected).

**Figure 2 molecules-27-00849-f002:**
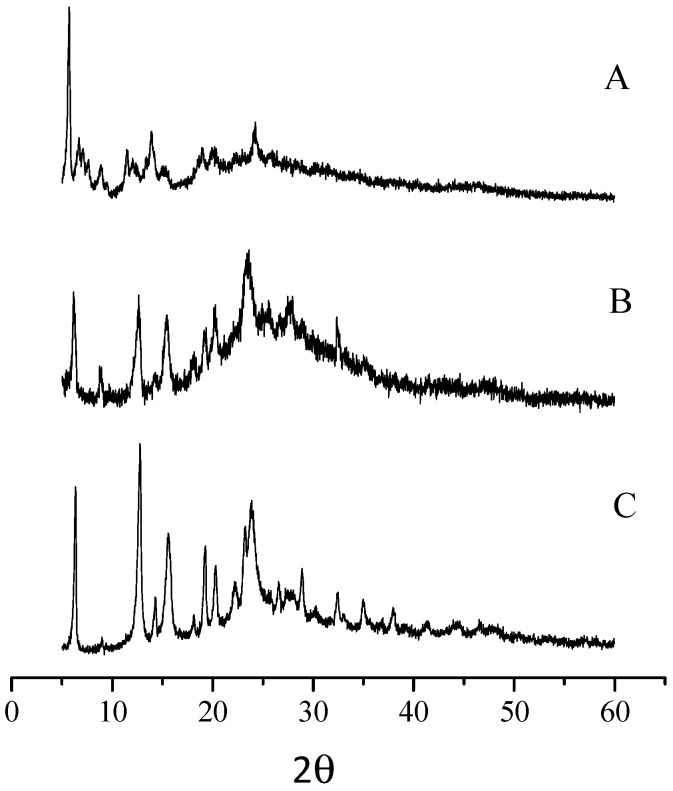
X-ray powder spectra of [Py_8_TPyzPzIn(OAc)]∙8H_2_O (**A**), [Py_8_TPyzPzAlCl]∙8H_2_O (**B**) and [Py_8_TPyzPzGaCl]∙4H_2_O (**C**).

**Figure 3 molecules-27-00849-f003:**
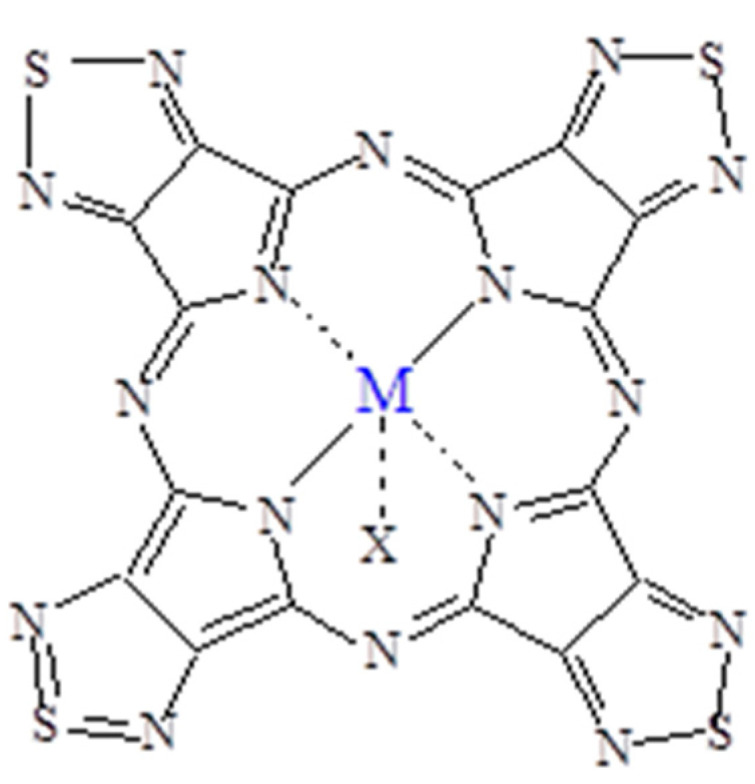
Schematic view of the complexes [TTDPzMX] (M = Al^III^, Ga^III^; X = Cl; M = In^III^, X = OAc) [5].

**Figure 4 molecules-27-00849-f004:**
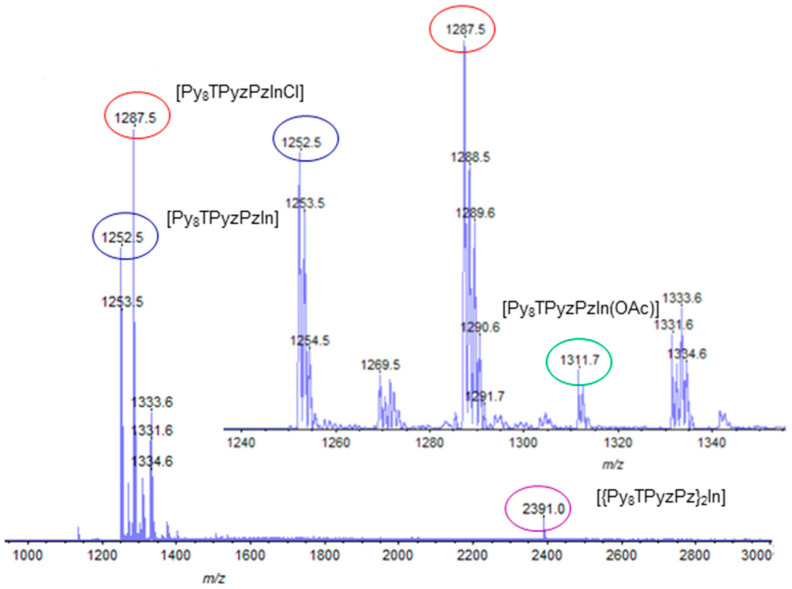
Mass spectrum of the complex [Py_8_TPyzPzIn(OAc)]·8H_2_O. The spectrum in the inset shows the expanded range *m*/*z* 1240–1350.

**Figure 5 molecules-27-00849-f005:**
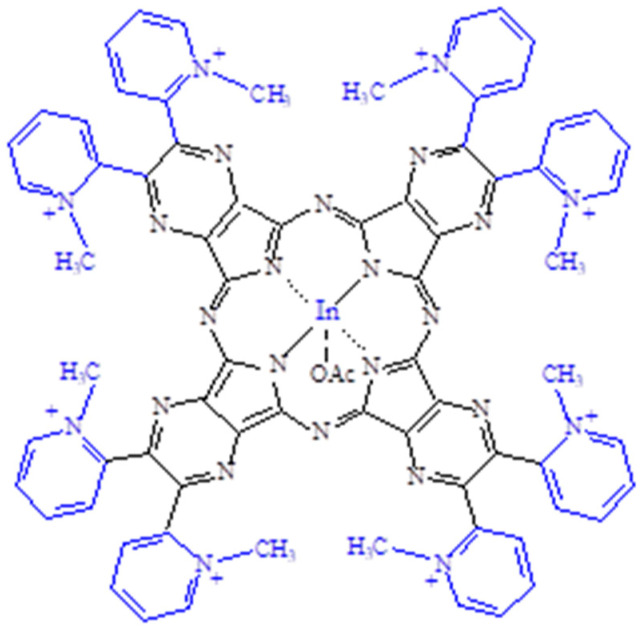
Schematic view of porphyrazine octacation [(2-Mepy)_8_TPyzPzIn(OAc)]^8+^ (salted by I^−^ anions; water neglected).

**Figure 7 molecules-27-00849-f007:**
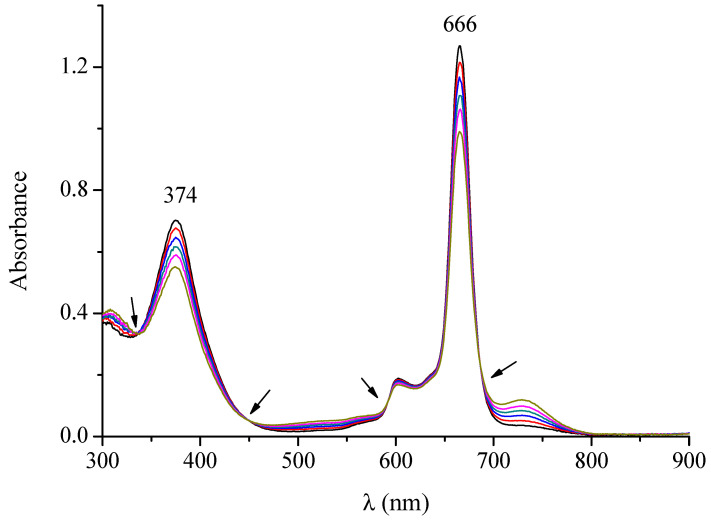
UV-visible spectral behavior in DMF of the complex [Py_8_TPyzPzIn(OAc)]·8H_2_O in the range 300–900 nm (arrows indicate isosbestic positions).

**Figure 8 molecules-27-00849-f008:**
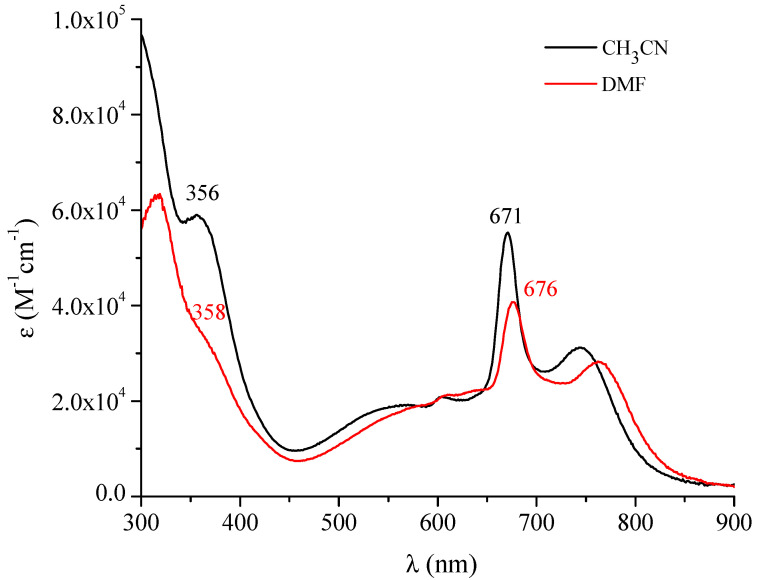
UV-visible spectra in the range 300–900 nm of the complex [(2-Mepy)_8_TPyzPzIn(OAc)](I)_8_·H_2_O in DMF and CH_3_CN.

**Figure 9 molecules-27-00849-f009:**
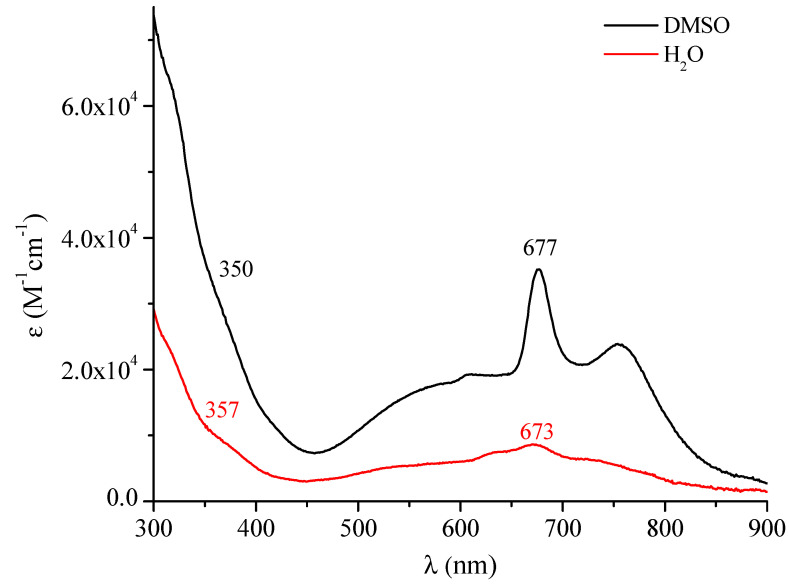
UV-visible spectra in the range 300–900 nm of the complex. [(2-Mepy)_8_TPyzPzIn(OAc)](I)_8_·H_2_O in DMSO and H_2_O.

**Figure 10 molecules-27-00849-f010:**
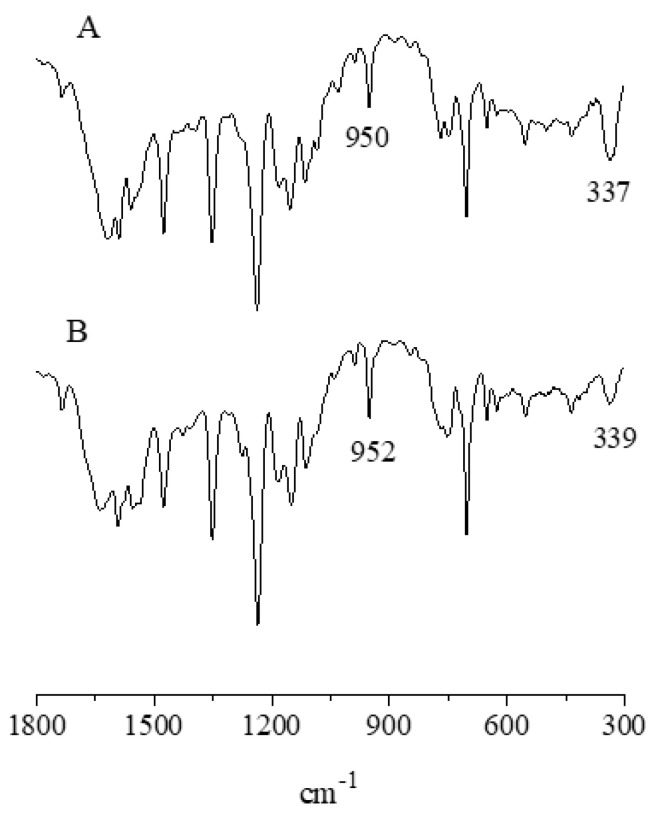
IR spectra in KBr of [(PdCl_2_)_4_Py_8_TPyzPzIn(OAc)]·8H_2_O (**A**) and [(PtCl_2_)_4_Py_8_TPyzPzIn(OAc)]·H_2_O (**B**).

**Figure 11 molecules-27-00849-f011:**
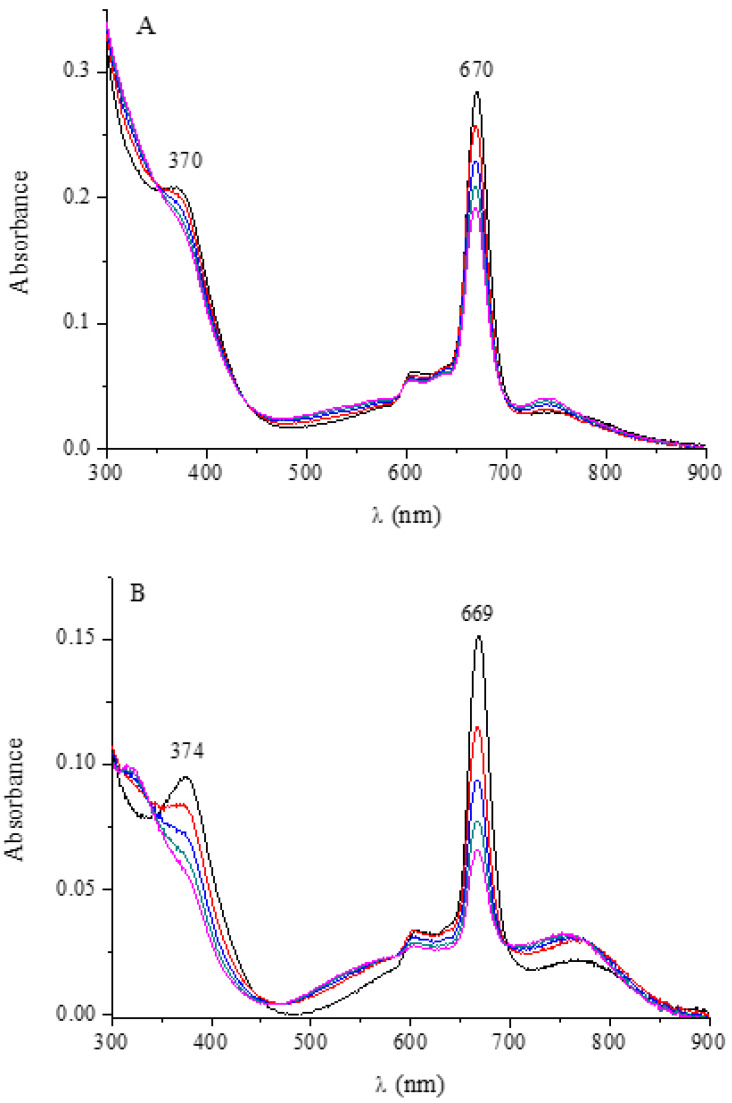
UV-visible spectra in the range 300–900 nm of [(PdCl_2_)_4_Py_8_TPyzPzIn(OAc)]·8H_2_O (**A**) and [(PtCl_2_)_4_Py_8_TPyzPzIn(OAc)]·H_2_O (**B**) in DMSO.

**Figure 12 molecules-27-00849-f012:**
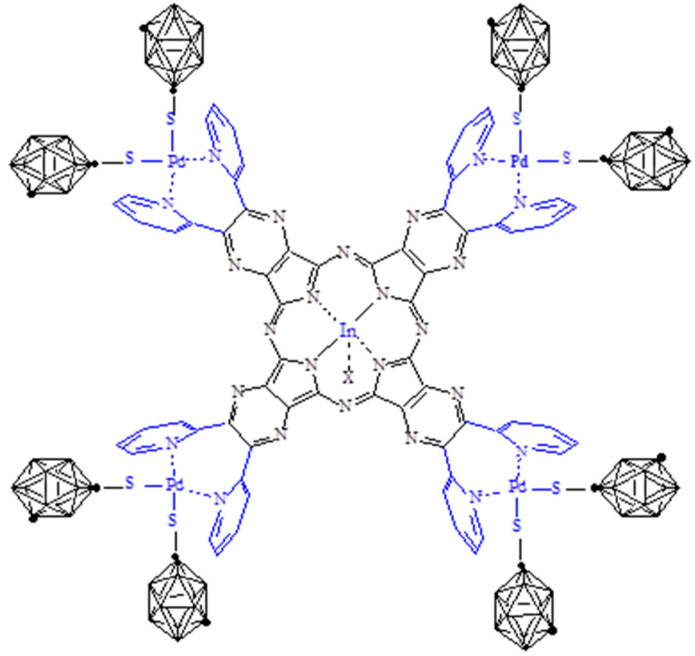
Schematic representation of the complex [{Pd(CBT)_2_}_4_Py_8_TPyzPzIn(OAc)]·19H_2_O (X = OAc^−^).

**Figure 13 molecules-27-00849-f013:**
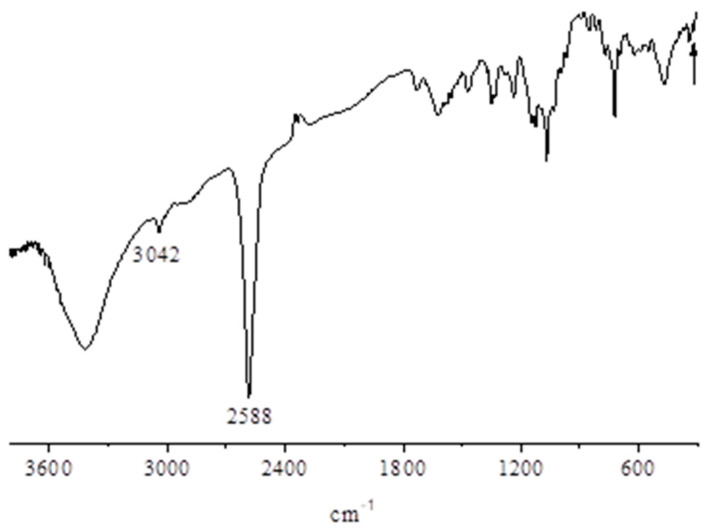
IR spectrum in KBr of the complex [{Pd(CBT_2_)}_4_Py_8_TPyzPzIn(OAc)]·19H_2_O.

**Figure 14 molecules-27-00849-f014:**
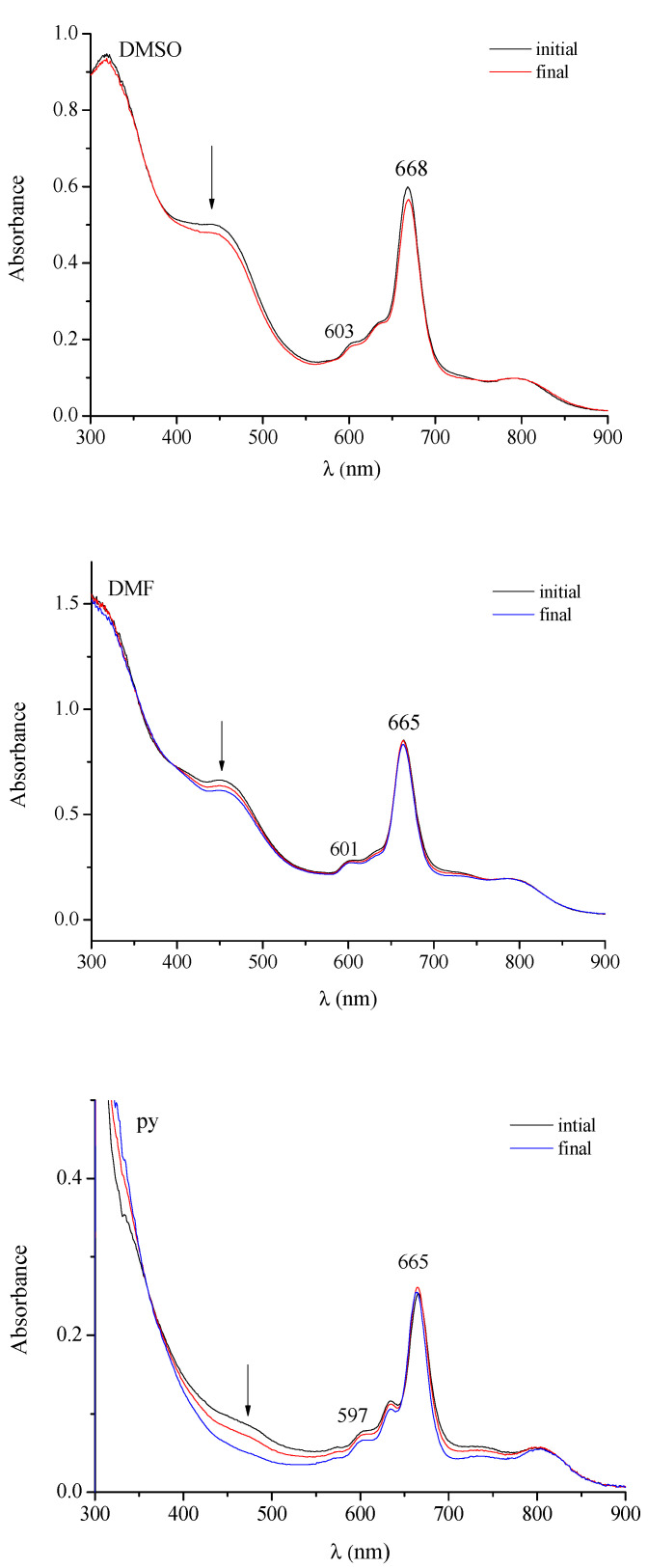
UV-visible spectra in the range 300–900 nm of the complex [{Pd(CBT_2_)}_4_Py_8_TPyzPzIn(OAc)]·19H_2_O in different solvents.

**Figure 15 molecules-27-00849-f015:**
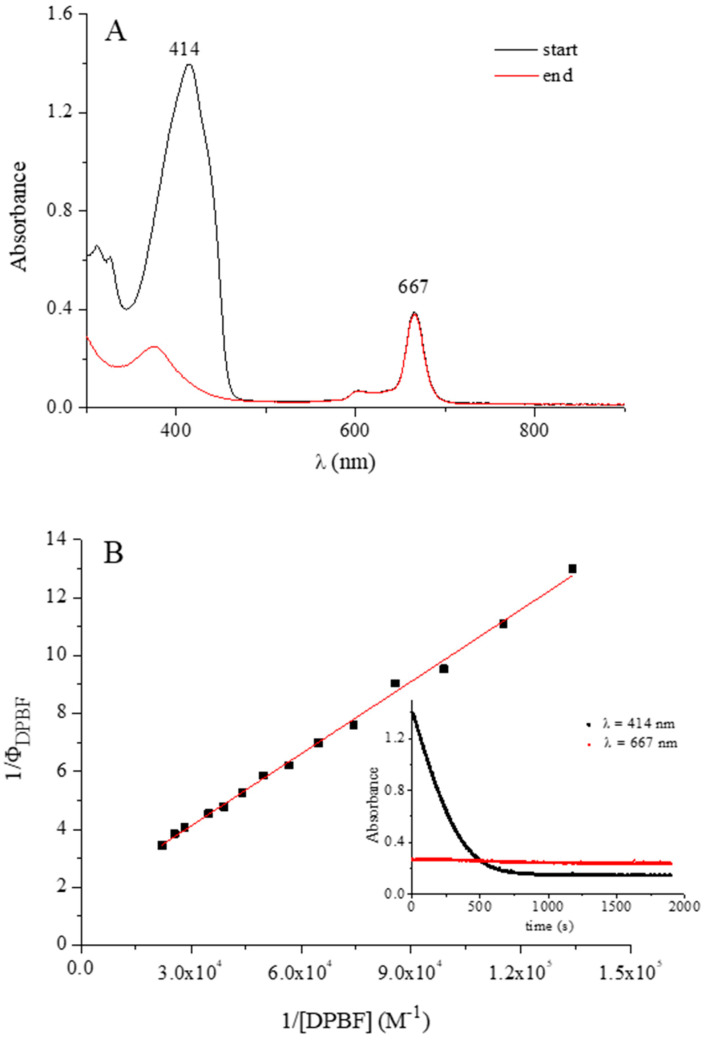
(**A**) UV-visible spectra in DMF solution of [Py_8_TPyzPzIn(OAc)] and DPBF before (black line) and after (red line) laser irradiation (λ_irr_ = 670 nm). (**B**) Stern-Volmer data analysis of the DPBF photooxidation (DPBF absorption decay at 414 nm is shown in the inset; the red dotted line indicates stability of the complex during irradiation).

**Table 1 molecules-27-00849-t001:** UV-visible spectral data (λ, nm (logε)) of [Py_8_TPyzPzIn(OAc)], [(2-Mepy)_8_TPyzPzIn(OAc)](I)_8_, [(MCl_2_)_4_Py_8_TPyzPzIn(OAc)] (M = Pd^II^, Pt^II^) and [{Pd(CBT)_2_}_4_Py_8_TPyzPzIn(OAc)] in different solvents (specific presence of water evidenced inside the table).

Macrocyclic Species	Solvent	Soret Region		Q-Band Region		
[Py_8_TPyzPzIn(OAc)]·8H_2_O	CH_3_CN	371 (4.97)		599 (4.36)	662 (5.21)	
	CHCl_3_	371 (4.99)		600 (4.42)	662 (5.29)	
	DMF	374 (4.97)		603 (4.19)	666 (5.22)	
	DMSO	376 (4.97)		603 (4.40)	667 (5.26)	
	Py	378 (5.01)		606 (4.41)	669 (5.26)	
[(2-Mepy)_8_TPyzPzIn(OAc)](I)_8_·H_2_O	CH_3_CN	356 (4.77)	560 (4.28)	603 (4.32)	671 (4.74)	745 (4.50)
	DMF	314 (4.80) 358(sh) (4.55)	575 (4.27)	609 (4.33)	676 (4.61)	763 (4.45)
	DMSO	315(sh) (4.81) 350 (sh) (4.56)	577 (4.25)	608 (4.28)	677 (4.55)	755 (4.38)
	H_2_O	357(sh) (4.02)	570 (3.74)		636 (3.87) 673 (3.93)	719 (2.81)
	Py	357(sh) (4.32)	573 (3.93)	611 (4.00)	681 (4.27)	771 (4.09)
[(PdCl_2_)_4_Py_8_TPyzPzIn(OAc)]·8H_2_O	DMSO	377 (4.37)		603 (4.07)	672 (4.74)	
	DMF	366 sh (4.52)		608 (4.08)	672 (4.61)	
	Py	377 (4.59)		605 (4.10)	668 (4.79)	
	CH_3_CN				673	
[(PtCl_2_)_4_Py_8_TPyzPzIn(OAc)]·H_2_O	DMSO	374 (4.48)		604 (4.03)	669 (4.69)	
	DMF	373 (4.65)		603 (4.20)	667 (4.83)	
	Py	378 (4.57)		607 (4.12)	668 (4.77)	
[{Pd(CBT)_2_}_4_Py_8_TPyzPzIn(OAc)]∙19H_2_O	DMSO	320 (4.84)	454 (4.52)	603 (3.98)	668 (4.42)	
	DMF	318 (4.91)	454 (4.60)	603 (4.15)	666 (4.50)	
	Py	320 (4.94)	468 (4.43)	601 (4.27)	666 (4.56)	

**Table 2 molecules-27-00849-t002:** Singlet Oxygen Quantum Yields (Φ_Δ_) in DMF (or DMF/HCl) of [Py_8_TPyzPzIn(OAc)], [(MCl_2_)_4_Py_8_TPyzPzIn(OAc)] (M = Pd^II^, Pt^II^) and Al^III^ and Ga^III^ analogs.

Compound	HCl [M]	λ_max_ [nm]	λ_irr_ [nm]	Φ_Δ_ ^a^	Ref.
[Py_8_TPyzPzIn(OAc)]	0	667	670	0.55	tw
[(PdCl_2_)_4_Py_8_TPyzPzIn(OAc)]	0	672	670	0.36	tw
[(PtCl_2_)_4_Py_8_TPyzPzIn(OAc)]	0	667	670	0.46	tw
[Py_8_TPyzPzAlCl]	1 × 10^−4^	656	660	0.24	7
[Py_8_TPyzPzGaCl]	1 × 10^−4^	652	650	0.68	7
[(PdCl_2_)_4_Py_8_TPyzPzAlCl]	1 × 10^−4^	662	660	0.21	7
[(PdCl_2_)_4_Py_8_TPyzPzGaCl]	1 × 10^−4^	656	660	0.42	7

^a^ Mean value of at least three measurements. Uncertainty is half dispersion and it is typically ±0.03 under the utilized experimental conditions (water molecules neglected).

## Data Availability

Not applicable.

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
