# Peer review of "Tetra-2,3-Pyrazinoporphyrazines with Externally Appended Pyridine Rings 22 Synthesis, Physicochemical and Photoactivity Studies on In(III) Mono- and Heteropentanuclear Complexes"

_molecules, 2022, doi:10.3390/molecules27030849_

Round 1

Reviewer 1 Report

The article can be published after the following suggestions:

1- The abstract section must be clear and informative to can any reader know what the authors do in this article. So, it must be re-written.

2- The introduction section must be re-written also, this is include a little review and only contain about three references, so, it must be written in improved state.

3- Why the authors do not make 1HNMR analysis? 

4- I suggest that the better to do crystal structure analysis to sure the structure of synthesized compound.

5- The figures and graphical figures must be improved to become clear view.  

Author Response

Reviewer 1

1) The abstract section must be clear and informative to can any reader know what the authors do in this article. So, it must be rewritten.

 Abstract adequately modified.

2) The introduction section must be re-written also, this is include a little review and only contain about three references, so, it must be written in improved state.

Introduction changed in the direction suggested by the reviewer.

3) Why the authors do not make 1HNMR analysis?

NMR believed not strictly necessary. In fact, in our previous extraordinary numerous published reports (21): a) always reproducibly verified that synthetic work leads in all cases to the exclusive formation of the anionic macrocyclic porphyrazine anion [Py8TPyzPz]2- as also confirmed in some of the reports by detailed NMR spectral results (Inorg. Chem. 2004, 43, 8626-8636; IC, 2008, 47, 3903-3919; IC, 2010, 49, 2447-2456; IC, 2011, 50, 7391-7402).

4) I suggest that the better to do crystal structure analysis to sure the structure of synthesized compound.

All the In(III) reported complexes are obtained as having poor crystallìne character due to their tendency to retain water molecules. All our attempts to obtain crystals suitable for X-ray analysis were unsuccessful.

5) The figures and graphical figures must be improved to become clear view.

Specifically Figures 3, 4, 5, 6, 7, 11,13 and Graphical abstract have been replaced with improved figures. 

Reviewer 2 Report

The manuscript from Donzello et al. reports the synthesis and the physicochemical characterization of five In(III)-porphyrazine complexes as completion of the previously reported Al(III) and Ga(III) analogues. Consequently, I consider the lack of novelty one of the main problems and reasons to reject the present article.

Furthermore:

1) The introduction does not explain the research context to which those complexes refer nor their research interest. The synthetic choices are also not justified. The sentence “It has been thought that this latter engagement on the tervalent metal ion derivatives required to be appropriately fulfilled with the obvious extension to the parallel work on the related series of In(III) derivatives” is not acceptable.

2) Self-citation (notably in the introduction) is not acceptable.

3) The experimental values for the elemental analysis are different (sometimes too different) from the theoretical ones bringing serious doubts on the purity of the complexes and, consequently, on the correctness of the UV/Vis spectra and the authors interpretation.

4) Figure 4 shows two mass spectra which were probably manually modified. It seems that a white line covers some background noise.

Author Response

Reviewer 2

Main comment: The manuscript from Donzello et al. reports the synthesis and the physicochemical characterization of five In(III)-porphyrazine complexes as completion of the previously reported Al(III) and Ga(III) analogues. Consequently, I consider the lack of novelty one of the main problems and reasons to reject the present article.

As to the proposed rejection of the manuscript in the main comment of the reviewer, the authors notice that the new manuscript reports on mononuclear neutral and  octacationic macrocycles and on related heteropentanuclear porphyrazine macrocycles all of them examined in detail as potential curative anticancer agents. The criterium of the given content to this number 22 report is specific as was that used in the related previous 21, 19 of them published in the “Inorganic Chemistry” journal.

1) The introduction does not explain the research context to which those complexes refer nor their research interest. The synthetic choices are also not justified. The sentence “It has been thought that this latter engagement on the tervalent metal ion derivatives required to be appropriately fulfilled with the obvious extension to the parallel work on the related series of In(III) derivatives” is not acceptable.

The introduction has been adequately modified, This all reveiwer’s comment duplicates what written by the reviewer in the main comment and appears to the authors too severe.

2) Self-citation (notably in the introduction) is not acceptable.

Self-citation is restricted to what is extremely necessary.

3) The experimental values for the elemental analysis are different (sometimes too different) from the theoretical ones bringing serious doubts on the purity of the complexes and, consequently, on the correctness of the UV/Vis spectra and the authors interpretation.

First, the authors remark that the same macrocyclic body has been exclusively formed and dealt with in all the 21 previously published reports. Some different results only regard H analysis and this might depend on the fact that all the complexes retain water molecules which can give different results from sample to sample of the same species when brought to the instrument for examination. It should be noticed that in all cases the elemental analysis of In, Pd, and Pt metal centers, added on purpose, give in all cases perfect results. No doubts then about purity of the complexes and the correctness of the UV/Vis spectra and the authors interpretation.

4) Figure 4 shows two mass spectra which were probably manually modified. It seems that a white line covers some background noise.

Right observation: the Figure has been replaced with the correct one.

Reviewer 3 Report

This manuscript is well written for a preparation of a versatile ligand molecules with tunable terminal substructures for several metal ions including their coulter anions. The medicinal applications had achieved by conventional approaches the results were in the predictable range.  However, this reviewer feels that this manuscript is sufficiently worth to be published as a Molecules article after minor revision.

  • Molecular structures are complicated but are not presented sufficiently. The octacation species should be illustrated in Figure 1 or a relatively early part.
  • P 2, line 3 of Solvents and Reagent part. CaCarborane => carborane.
  • P3, Figure 3. The resolution of the figure is low. It should be replaced.
  • P4, line 5 of Results and Discussion part; P6, line 4. Experimental section => Experimental Section
  • P6, line 6. bexternal => external
  • P7, Figure 6; P8, Figure 7; P13, Figure 13. The resolutions of the figures are supposed to be low. It should be replaced if possible.
  • P11, line 3. i.e.sodium => i.e. sodium
  • P11, line 7-8. cis- and trans- => italic
  • P 11, Figure 11. The meta-carbons of the two carborane units at the bottom positions should be marked.
  • P16, line 14 of Conclusion part. PDT => (PDT)
  • P16, Supplementary material parts. The lower cases are not reflected for most molecular formula. Please check and modify.
  • P17, GA figure. The resolution of the molecular structure is extremely low. It must be replaced.

Author Response

Reviewer 3

  • Molecular structures are complicated but are not presented sufficiently. The octacation species should be illustrated in Figure1 or a relatively early part.

A schematic representation of the structure of the octacation has been added as Figure 5

  • P 2, line 3 of Solvents and Reagent part. CaCarborane =>carborane.

Corrected

  • P3, Figure 3. The resolution of the figure is low. It should bereplaced.

Figure 3 replaced

  • P4, line 5 of Results and Discussion part; P6, line 4.Experimental section => Experimental Section

           Correction made

  • P6, line 6. bexternal => external

Correction made

  • P7, Figure 6; P8, Figure 7; P13, Figure 13. The resolutions of thefigures are supposed to be low. It should be replaced if possible.

      Figures replaced

  • P11, line 3. i.e.sodium => i.e. sodium

          Correction made

  • P11, line 7-8. cis- and trans- => italic

Correction made

  • P 11, Figure 11. The meta-carbons of the two carborane units at the bottom positions should be marked.

           Figure corrected

  • P16, line 14 of Conclusion part. PDT => (PDT)

            Correction made

  • P16, Supplementary material parts. The lower cases are not reflected for most molecular formula. Please check and modify.

            Lower cases checked and modified

  • P17, GA figure. The resolution of the molecular structure isextremely low. It must be replaced.

          GA figure replaced

Round 2

Reviewer 1 Report

Now, the article can be accepted

Author Response

Many thanks to the reviewer 1 for his/her acceptance of the revised manuscript

Reviewer 2 Report

I confirm what previously written as no major changes have been made.

Furthermore, the updated version has some problems with the figures.

Author Response

The authors thank the reviewer 2 for his/her previous comments which, in author's opimion allowed to improve the manuscrpt,